# A method for image registration via broken geodesics

Alphin J Thottupattu[1], Jayanthi Sivaswamy[1], and Venkateswaran P. Krishnan[2]

[1] International Institute of Information Technology, Hyderabad 500032, India
alphinj.thottupattu@research.iiit.ac.in
[2] TIFR Centre for Applicable Mathematics, Bangalore 560065, India

**Abstract.** Anatomical variabilities seen in longitudinal data or inter-subject data is usually described by the underlying deformation, captured by non-rigid registration of these images. Stationary Velocity Field (SVF) based non-rigid registration algorithms are widely used for registration. However, these methods cover only a limited degree of deformations. We address this limitation and define an approximate metric space for the manifold of diffeomorphisms $\mathcal{G}$. We propose a method to break down the large deformation into finite set of small sequential deformations. This results in a broken geodesic path on $\mathcal{G}$ and its length now forms an approximate registration metric. We illustrate the method using a simple, intensity-based, log-demon implementation. Validation results of the proposed method show that it can capture large and complex deformations while producing qualitatively better results than state-of-the-art methods. The results also demonstrate that the proposed registration metric is a good indicator of the degree of deformation.

**Keywords:** Large Deformation · Inter-subject Registration · Approximate Registration Metric

## 1 Introduction

Computational anatomy is an area of research focused on developing computational models of biological organs to study the anatomical variabilities in the deformation space. Anatomical variations arise due to structural differences across individuals and changes due to growth or atrophy in an individual. These variations are studied using the deformation between the scans captured by a registration step. The registration algorithms typically optimize an energy functional based on a similarity function computed between the fixed and moving images. Many initial image registration attempts use energy functionals inspired by physical processes to model the deformation as an elastic deformation [11], or viscous flow [20] or diffusion [14]. The diffusion-based approaches have been explored for 3D medical images in general [6] and with deformations constrained to be diffeomorphic [23] to ensure preservation of the topology. The two main approaches used to capture diffeomorphisms are parametric and nonparametric methods. The Free Form Deformation (FFD) model [5,12] is a widely used parametric deformation model for medical image registration, where a rectangular grid with

control points is used to model the deformation. Large diffeomorphic deformations [12] are handled by concatenating multiple FFDs. Deformable Registration via Attribute Matching and Mutual-Saliency Weighting (DRAMMS) [7] is a popular FFD-based method, which also handles inter-subject registration. DRAMMS matches Gabor features and prioritizes the reliable matching between images while performing registration. The main drawback of the deformations captured by FFD models is that they do not guarantee invertibility. The non-parametric methods represent the deformation with stationary or time varying velocity vector field. The diffeomorphic log-demon [23] is an example of the former while the Large Deformation Diffeomorphic Metric Mapping (LDDMM) [15] inspired from [8] is an example of the latter approach. In LDDMM, deformations are defined as geodesics on a Riemannian manifold, which is attractive; however, the methods based on this framework are computationally complex. The diffeomorphic log-demon framework [23], on the other hand, assigns a Lie group structure and assumes a stationary velocity field (SVF) which leads to computationally efficient methods, which is of interest to the community for practical purposes. This has motivated the exploration of a stationary LDDMM framework [16] that leverages the SVF advantage. The captured deformations are constrained to be symmetric in time-varying LDDMM [1] and log-demon [21] methods. Choosing an efficient optimization scheme such as Gauss-Newton as in [10] reduces the computational complexity of LDDMM framework. However, the log-demon framework is of interest to the community for practical purposes because of its computational efficiency and simplicity.

The Lie group structure gives a locally defined group exponential map to map the SVF to the deformation. Thus log-demon framework is meant to capture only neighboring elements in the manifold, i.e., only a limited degree of deformations can be captured. This will be referred to as the limited coverage issue of the SVF methods in this paper. Notwithstanding the limited coverage, several SVF based methods have been reported for efficient medical image registration with different similarity metrics, sim, such as local correlation between the images [17], spectral features [3], modality independent neighborhood descriptors [19] and wavelet features [18,9].

SVF based algorithms cannot handle complex deformations because the deformations are constrained to be smooth for the entire image and thus constrain the possible degree of deformation to some extent. We address this drawback by splitting the large deformation into finite set of smaller deformations. The key contributions of the paper are: i) an SVF-based registration framework to handle large deformations such as inter-subject variations computationally efficiently ii) an approximate metric to quantify structural variations between two images.

## 1.1 Background

Let $G$ be a finite-dimensional Lie group with Lie algebra $\mathfrak{g}$. Recall that $\mathfrak{g}$ is the tangent space $T_e G$ at the identity $e$ of $G$. The exponential map $\exp : \mathfrak{g} \to G$ is defined as follows: Let $v \in \mathfrak{g}$. Then $\exp(v) = \gamma_v(1)$, where $\gamma$ is the unique one-parameter subgroup of the Lie group $G$ with $v$ being its tangent vector at

$e$. The vector $v$ is called the infinitesimal generator of $\gamma$. The exponential map is a diffeomorphism from a small neighborhood containing 0 in the Lie algebra $\mathfrak{g}$ to a small neighborhood containing $e$ of $G$.

Due to the fact that a bi-invariant metric may not exist for most of the Lie groups considered in medical image registration, the deformations considered here are elements of a Lie group with the Cartan-Schouten Connection [24]. This is the same as the one considered in the log-demon framework [23]. This is a left invariant connection [22] in which geodesics through the identity are one-parameter subgroups. The group geodesics are the geodesics of the connection. Any two neighboring points can be connected with a group geodesic. That is, if the stationary velocity field $v$ connecting two images in the manifold $\mathcal{G}$ is small enough, then its group exponential map forms a geodesic. Similarly every $\mathfrak{g} \in G$ has a geodesically convex open neighbourhood [22].

## 2   Method

SVF based registration methods capture only a limited degree of deformation because exponential mappings are only locally defined. In order to perform registration of a moving image towards a fixed image, SVF is computed iteratively by updating it with a smoothed velocity field. This update is computed via a similarity metric that measures the correspondence between the moving and fixed images. The spatial smoothing has a detrimental effect as we explain next. A complex deformation typically consists of spatially independent deformations in a local neighbourhood. Depending on the smoothing parameter value, only major SVF updates in each region is considered for registration. Thus, modeling complex deformations with a smooth stationary velocity field is highly dependent on the similarity metric and the smoothing parameter in a registration algorithm. Finding an ideal similarity metric and an appropriate smoothing parameter applicable for any registration problem, irrespective of the complexity of the deformation and the type of data, is difficult.

We propose to address this issue as follows: Deform the moving image toward the fixed image by sequentially applying an SVF based registration. The SVF based algorithm chooses the major or the predominant (correspondence-based) deformation component among the spatially independent deformations in all the neighbourhoods to register along these predominant directions. The subsequent steps in the algorithm captures the next set of predominant directions sequentially. These sequentially captured deformations has a decreasing order of degree of pixel displacement caused by the deformations. Mathematically speaking, the discussion above can be summarised as follows. Consider complex deformations as a set of finite group geodesics and use a registration metric approximation to quantify the deformation between two images in terms of the length of a broken geodesic connecting them; a broken geodesic is a piecewise smooth curve, where each curve segment is a geodesic.

In the proposed method, the similarity-based metric selects the predominant deformation in each sequential step. The deformation that can bring the moving

image in a step maximally closer to the target is selected from the one-parameter subgroup of deformations. In the manifold $\mathcal{G}$ every geodesic is contained in a unique maximal geodesic. Hence the maximal group geodesic $\gamma_i$ computed using log-demon registration framework deforms the sequential image $S_{i-1}$ in the previous step maximally closer to $S_N$. The maximal group geodesic paths are composed to get the broken geodesic path. As the deformation segments are diffeomorphic, the composed large deformation of the segments also preserves diffeomorphism to some extent.

In the proposed method, the coverage of the SVF method and the degree of deformation determines the number of subgroups $N$ needed to cover the space. The feature based SVF methods in general, give more coverage for a single such subgroup and reduce the value of $N$.

A broken geodesic $\gamma : [0, T] \to M$ has finite number of geodesic segments $\gamma_i$ for partitions of the domain $0 < t_1 < t_2 < \cdots < t_i < \cdots t_N = T$ where $i = 1, ....N$. The proposed algorithm to deform $S_0$ towards $S_N$ is given in Algorithm 1. We have chosen the registration algorithm from [21] to compute SVF, $u_i$, in Algorithm 1. The Energy term is defined as: $\text{Energy}(S_i, S_N) = \text{sim}(S_N, S_i) + \text{Reg}(\gamma_i)$ where the first term is a functional of the similarity measure, which captures the correspondence between images, with $\text{sim}(S_N, S_i) = S_N - S_{i-1} \circ \exp(v_i)$. The second term is a regularization term, with $\text{Reg}(\gamma_i) = \|\nabla \gamma_i\|^2$.

---

**Algorithm 1** Proposed Algorithm

---

1: Input: $S_0$ and $S_N$
2: Result: Transformation $\gamma = \exp(v_1) \circ \exp(v_2) \circ ... \exp(v_N)$
3: Initialization: $E_{\min} = \text{Energy}(S_0, S_N)$
4: **repeat**
5:     Register $S_{i-1}$ to $S_N \to u_i$
6:     Temp $= S_{i-1} \circ \exp(u_i)$
7:     $E_i = \text{Energy}(\text{Temp}, S_N)$
8:     **if** $E_i < E_{\min}$ **then**
9:         $v_i = u_i$
10:        $E_{\min} = E_i$
11:        $S_i = \text{Temp}$
12:     **end if**
13: **until** Convergence

---

### 2.1 Registration metric approximation

Let $\gamma$ be a broken geodesic decomposed into $N$ geodesics $\gamma_i$ with stationary field $v_i$, i.e. $\dot{\gamma}_i = v_i(\gamma(t)) \in T_{\gamma_i(t)}M$. Each of the constant velocity paths $\gamma_i$ is parameterized by the time interval $[t_{i-1}, t_i]$, and $N \in \mathbb{N}$ is minimized by requiring each of the geodesics in the broken geodesic to be maximal geodesics. The length

of the broken geodesic is defined as,

$$l(\gamma) = \sum_i^N l(\gamma_i) = \sum_i^N d(S_{i-1}, S_i) \qquad (1)$$

where, $d$ is a distance metric defined in Equation 2 .

$$d(S_{i-1}, S_i) = \inf\{\|v_i\|_V, S_{i-1} \circ \exp(v_i) = S_i\}. \qquad (2)$$

A registration metric needs to be defined to quantify the deformation between two images. The shape metric approximation in [25] can be used for the group geodesics of the Cartan-Schouten connection defined in the finite dimensional case as no bi-invariant metric exists. The length of a broken geodesic $l(\gamma)$ on the manifold $\mathcal{G}$ connecting $S_0$ and $S_N$, computed by Equation 1 is defined as the proposed approximate metric.

## 3   RESULTS

The proposed method was implemented using a simple intensity based log-demon technique [4] for illustrating the concept which is openly available at: `http://dx.doi.org/10.17632/29ssbs4tzf.1`. This choice also facilitates understanding the key strengths of the method independently. Two state-of-the-art (SOTA) methods are considered for performance comparison with the proposed method: the symmetric LDDMM implementation in ANTs [1] and DRAMMS which is a feature based, free-form deformation estimation method [7]. These two methods are considered to be good tools for inter-subject registration [26]. Publicly available codes were used for the SOTA methods with parameter settings as suggested in [26] for optimal performance. Both methods were implemented with B-spline interpolation, unless specified. 3D registration was done, and the images used in the experiments are 1.5T T1 MRI scans sourced from [2] and [13] unless specified otherwise. The number of maximum pieces in the broken geodesic path is set as five in all the experiments. The proposed image registration algorithm was used to register MRIs of different individuals.

### 3.1   Visual Assessment of Registration

To analyse the performance visually, six 3T MRI scans were collected. Three images collected from 20-30 year old male subjects were considered as moving images and three images collected from 40-50 year old female subjects were considered as fixed images. Performing a good registration is challenging with this selection of moving and fixed images. The high resolution MRI scans used for this experiment are openly available at `http://dx.doi.org/10.17632/gnhg9n76nn.1`. The registration results for these three different pairs are shown in Fig.1 -A. where only a sample slice is visualized for the 3 cases. The quality of registration

can be assessed by observing the degree of match between images in the last two rows of each column. The mean squared error (MSE) was used as a similarity metric along with cubic interpolation. The results indicate that the proposed method is good at capturing complex inter-subject deformations.

The performance of the proposed method on medical images was compared with the state-of-the-art methods in Fig.1 -B. To apply the computed deformation, linear interpolation was used in all the methods. ANTs and the proposed method used MSE as a similarity metric for fair comparison and DRAMMS used its Gabor feature-based metric as it is a feature based method. The results shows that the deformations at the sulcal regions are better captured by the proposed method.

The quality of inverted deformations captured with ANTs and proposed method were also compared as follows. In Fig.1 -C the moving image deformed with moving-fixed deformation and fixed image deformed with inverted moving-fixed deformation are analysed for both the methods. The arrows overlaid on the registered images highlight regions where the proposed method yields error-free results as opposed to the other method. The results with proposed method shows better visual similarity with the target images in each case.

### 3.2   Quantitative Assessment of Registration

We present a quantitative comparison of the proposed method compared with ANTs and DRAMMS under the same setting. The average MSE for 10 image pair registrations with ANTs was $0.0036 \pm 0.0009$, with DRAMMS it was $0.0113 \pm 0.0068$ and with the proposed method it was $0.0012 \pm 7.0552e - 08$.

The computed deformations in each method were used to transfer region segmentation (labels) from the moving image to the fixed image. The transferred segmentations are assessed using the Dice metric. Fig.2 shows a box plot of the obtained Dice values calculated by registering 10 pairs of brain MRIs with the fixed image, for white matter (WM), grey matter (GM) and 2 structures (L & R-hippocampus). The segmentation results for larger structures (i.e., WM and GM) are better with the proposed method compared to the other methods, whereas the smaller structure segmentation is comparable to DRAMMS.

### 3.3   Validation of proposed registration metric

Finally, a validation of the proposed registration metric was done using two age-differentiated (20-30 versus 70-90 years) sets of MRIs, of 6 female subjects. Images from these 3D image sets were registered to an (independently drawn) MRI of a 20 year-old subject. The proposed registration metric was computed for the 6 pairs of registrations. A box plot of the registration metric value for each age group is shown in Fig.3. Since the fixed image is that of a young subject, the registration metric value should be higher for the older group than for the younger group, which is confirmed by the plot. Hence, it can be concluded that the proposed registration metric is a good indicator of natural deformations.

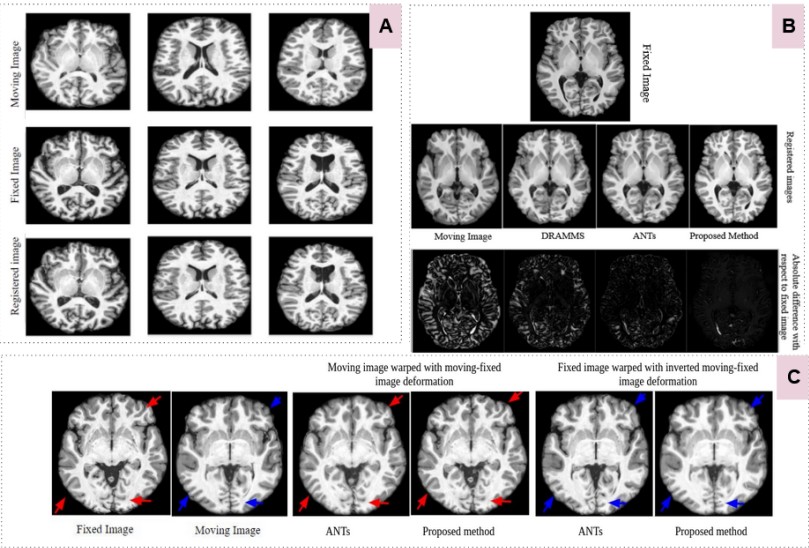

**Fig. 1.** A) Inter-subject image registration with proposed method for 3 pairs of volumes (in 3 columns) using cubic interpolation. Only sample slices are shown. B) Inter-subject image registration with 3 methods: DRAMMS, ANTs and the proposed method, implemented with linear interpolation. The regions near same colour arrows can be compared to check the registration accuracy. C) Forward and Backward Image Registration. Blue (Red) arrow shows where proposed method yields error-free results in moving (fixed) images, fixed (moving) images and warped moving (fixed) image using moving-fixed(inverted moving-fixed) deformation. Inverted moving-fixed deformation applied on fixed image and proposed method captures finer details compared to ANTs.

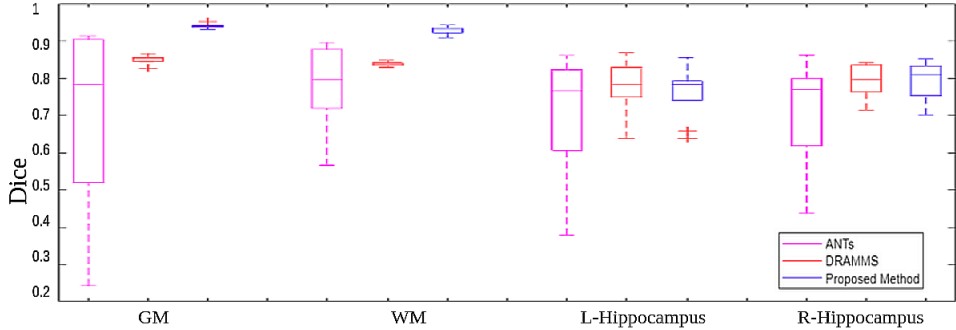

**Fig. 2.** Assessment of registration via segmentation of different structures using ANTS (magenta), DRAMMS (red), and the proposed method (blue). Box plots for the Dice coefficient are shown for White Matter (WM), Gray Matter (GM) and the Left and Right Hippocampi.

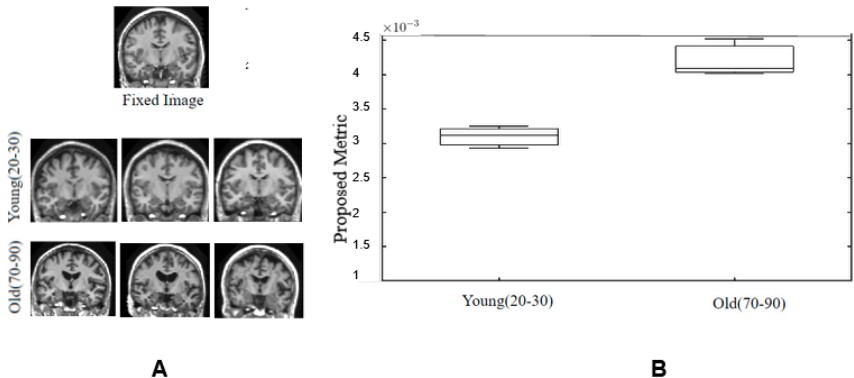

**Fig. 3.** Validation of the proposed registration metric.A) Central slices of images used to perform registration B) Box plots of the proposed registration metric values for registration of the fixed image with images of young and old subject group.

## 4  Discussion

Group exponential map based methods, with simple similarity registration metrics, fail to capture large deformations as the map is local in nature. We have addressed this issue in this paper by modelling large deformations with broken geodesic paths with the path length taken to be the associated registration metric. From the experiments it is observed that five pieces in the broken geodesic path is enough to capture very complex deformations. The proposed method does not guarantee diffeomorphism in a strict mathematical sense of infinite differentiability as the paths are modelled as piecewise geodesics. However, the experiments we have done suggest that the proposed method produces diffeomorphic paths. The results of implementation with a simple log-demon method show the performance to be superior to SOTA methods for complex/large deformations. We plan to extend this work by implementing the proposed framework using more efficient SVF based approaches such as in [3,17,19,18,9]. In summary, we have proposed a SVF-based registration framework that can capture large deformations and an approximate metric to quantify the shape variations between two images using the captured deformations.

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
