# OpenReview forum: "A method for image registration via broken geodesics"
_WBIR.info/2022/Workshop/Biomedical_Imaging_Registration — WBIR 2022_

### Official Review · Reviewer_n8Mu · 2022-02-19

**Rating:** 3
**Confidence:** 3

**Deanonymize Review:**

no

**Detailed Comments:**

1. Authors are dealing with large and complex deformations. Diffeomorphism preservation is a critical aspect in such scenarios. Showing Jacobian determinants of the estimated transformation field would help to understand both the complexity of deformation and how well the proposed algorithm preserves the diffeomorphism.

2. In Algo 1, the criteria of convergence are missing. What happens if the condition at line number 8 does not satisfy. How do we get $S_i$ for the next step?

3. Does the proposed algorithm free of hyperparameters? The discussion on hyperparameters is missing.

4. The difference image between fixed and registered images could be a suitable alternative to highlight the performance of a registration method.

5. Fig.1: With the proposed method, the registered image looks almost similar to the fixed image. I am a little skeptical. In the moving image, there are certain regions that look quite different from the fixed image. After registration, it seems that the proposed algorithm is creating new material in the moving image to match exactly with the fixed image. Numerically though that is possible but physically that is not realistic.


**Paper Type:**

both

**Strengths Weaknesses:**

The authors proposed an inter-subject registration method to deal with large and complex deformation. They suggested breaking down the large deformations into a finite set of small sequential deformations. Conceptually, the approach makes sense, but, for large and complex deformations the preservation of diffeomorphism needs to be handled carefully. Even authors point out in the paper that their method preserves diffeomorphism to some extent, but they have not explored that direction theoretically and numerically.

The paper is overall well written especially the introduction section.

Though the registered images look fascinating at a first glance, there is a little bit of skepticism. Please see the comments for details.

---

### Official Review · Reviewer_2pqS · 2022-02-19

**Rating:** 2
**Confidence:** 4
**Recommendation:** Short Oral

**Deanonymize Review:**

no

**Detailed Comments:**

Minor: The aspect ratio of the figures was variable, and would need to be addressed. Fig 2 is overly pixelated and should be embedded as a pdf or postscript.

Summary: This paper proposes the use of a set of SVFs for flexible diffeomorphic registration, which provides an approximate geodesic method. The work seems reasonably sound, but has limited evaluation and minimal novelty and fails to discuss several related works.

**Paper Type:**

methodological development

**Strengths Weaknesses:**

Strengths: A mathematical justification for SVF is given, and the motivation for composing several transformations, and their use as an approximate metric seems sounds. Both qualitative and quantitative results are provided, with an illustrative example in fig 3 showing how this can be used as a metric.

Weaknesses: Why were Geodesic shooting methods, e.g. [1] not discussed. These provide a more efficient alternative to LDDMM, which is more flexible than SVF. The relationship of the proposed method to the established ML technique of boosting was not discussed. This work also ignores all the recent work on learning SVFs, e.g. in voxelmorph [2], which may be better suited than a Demons based framework. This work also seems more similar to the Diffeomorphic demons [3] rather than log-demons, and the difference with this work should be discussed It wasn’t clear how the data were segmented and how much hyper-parameter optimisation was performed for each of the methods. Does the proposed method not run the risk of overfitting to the similarity term due to the “fluid” type registration? The convergence criteria is not described, and it would be helpful to evaluate possible issues with allowing large numbers of transformations.

[1] Ashburner, John, and Karl J. Friston. "Diffeomorphic registration using geodesic shooting and Gauss–Newton optimisation." NeuroImage 55.3 (2011): 954-967.
[2] Balakrishnan, Guha, et al. "VoxelMorph: a learning framework for deformable medical image registration." IEEE transactions on medical imaging 38.8 (2019): 1788-1800.
[3] Vercauteren, Tom, et al. "Diffeomorphic demons: Efficient non-parametric image registration." NeuroImage 45.1 (2009): S61-S72.

---

### Official Review · Reviewer_gDyU · 2022-02-19

**Rating:** 4
**Confidence:** 3

**Deanonymize Review:**

no

**Detailed Comments:**

- It would be useful to provide more details of the dataset chosen for evaluation
- What is the rationale for the number of maximum pieces in the broken geodesic path being set to 5?
- For Figure 2, additional statistics would be useful – what is the significance of the differences between the methods? For the left and right hippocampi, it seems as though the proposed method performed on par with DRAMMS, though for WM and GM the difference seems more significant. Can you comment on why this might occur?
- For Figure 3, the images of the slices could be removed and only the box plots shown – or perhaps split into two subfigures.
- Please make sure to label the y axes in Figures 2 and 3

Overall, this was an interesting paper with a thorough introduction and well thought out experiments to prove the robustness of the registration algorithm. A few improvements can be made in the rewriting of the methods section. The discussion could also be improved by including information concerning the limitations and future work.


**Paper Type:**

methodological development

**Strengths Weaknesses:**

Strengths:
- The introduction is very well written, and provides detailed information about previous work, motivation for the proposed method and the major contributions of the manuscript
- Sufficient experiments were performed to demonstrate the robustness of the proposed method

Weaknesses:
- Section 2.2 of the method in detail could be written more clearly in general– perhaps dividing this section into more paragraphs would help visually. Conceptually, the background details of the SVF could be included in the introduction, while the method section can focus on precisely what you have implemented to address the problems of using SVF based registration.
- What limitations does the proposed method have? Please detail these in the discussion.
- More details should be provided concerning future work, for instance, what additional experiments can be performed?

---

### Decision · Program_Chairs · 2022-02-22

Accept